# Genetic Variants in Patients with Multiple Head and Neck Paragangliomas: Dilemma in Management

**DOI:** 10.3390/biomedicines9060626

**Published:** 2021-05-31

**Authors:** Anasuya Guha, Ales Vicha, Tomas Zelinka, Zdenek Musil, Martin Chovanec

**Affiliations:** 1Department of Otorhinolaryngology, 3rd Faculty of Medicine, Charles University and University Hospital Kralovske Vinohrady, 100 34 Prague, Czech Republic; martin.chovanec@fnkv.cz; 2Department of Pediatric Hematology and Oncology, 2nd Faculty of Medicine, Charles University and University Hospital Motol, 150 06 Prague, Czech Republic; ales.vicha@fnmotol.cz; 3Department of Internal Medicine, 1st Faculty of Medicine, Charles University and General University Hospital, 128 08 Prague, Czech Republic; tomas.zelinka@vfn.cz; 4Department of Biology and Medical Genetics, 1st Faculty of Medicine, Charles University and General University Hospital, 128 00 Prague, Czech Republic; zdenek.musil@lf1.cuni.cz

**Keywords:** HNPGL, *SDHD* gene, *SDHB* gene, germline mutation, carotid body tumors, malignant paragangliomas, genetic counselling, incidentalomas

## Abstract

Multiple head and neck paragangliomas (HNPGLs) are neuroendocrine tumors of a mostly benign nature that can be associated with a syndrome, precipitated by the presence of a germline mutation. Familial forms of the disease are usually seen with mutations of *SDHx* genes, especially the *SDHD* gene. *SDHB* mutations are predisposed to malignant tumors. We found 6 patients with multiple tumors amongst 30 patients with HNPGLs during the period of 2016 to 2021. We discuss the phenotypic and genetic patterns in our patients with multiple HNPGLs and explore the management possibilities related to the disease. Fifty percent of our patients had incidental findings of HNPGLs. Twenty-one biochemically silent tumors were found. Four patients had germline mutations, and only one had a positive family history. Three out of five underwent surgery without permanent complications. Preventative measures (genetic counselling and tumor surveillance) represent the gold standard in effectively controlling the disease in index patients and their relatives. In terms of treatment, apart from surgical and radiotherapeutic interventions, new therapeutic measures such as gene targeted therapy have contributed very sparsely. With the lack of standardized protocols, management of patients with multiple HNPGLs still remains very challenging, especially in those with sporadic or malignant forms of the disease.

## 1. Introduction

Head and neck paragangliomas (HNPGLs) are rare slow-growing neuroendocrine tumors which can occur sporadically or as part of hereditary syndromes [1,2,3,4,5]. A systematic review identified 10 genes that can, if mutated, lead to HNPGLs (Table 1) [5]. These are the four subunits of the *succinate dehydrogenase* (*SDHx*) complex (e.g., *SDHA, SDHB, SDHC, SDHD*), *succinate dehydrogenase complex assembly factor 2* (*SDHAF2*), *von Hippel-Lindau* (*VHL*), *receptor tyrosine-protein kinase* (*RET*) *proto-oncogene*, *neurofibromatosis type I* (*NF1*), *transmembrane protein 127* (*TMEM127*) and the *hypoxia-induced factor 2 alpha* (*HIF-2α*) gene [3,4,5].

Amongst the four subunits of the *SDHx* and *SDHAF2* genes that lead to paraganglioma syndromes (PGL) types 1 to 5, the highest risk of developing head and neck tumors is seen in relation to *SDHD* (PGL1) and *SDHAF2* (PGL2) [5]. Patients with *SDHB* mutations frequently develop non-HNPGLs; that is, sympathetic PGLs in 52% to 84% and pheochromocytomas in 18% to 28% [6].

Only 31% associated with an *SDHB* mutation develop HNPGLs [6]. VHL and TMEM127 also have a high likelihood of developing pheochromocytomas [7]. Mutations in *SDHD* showed the highest risk for multiple PGLs (60–79%), whether synchronous or metachronous. The possibility of malignant HNPGLs is seen maximally with *SDHB* mutations [3,6,8], thus reducing the 5-year survival probability to 36% [9]. Multiple HNPGLs are found only in about 19% to 31% of patients afflicted with *SDHC*. PGL3 is relatively rare in comparison to PGLS1 and PGLS2, and the average age at diagnosis is also higher than for the other paragangliomas [10]. Multiplicity can also be seen, to a certain extent, with *SDHAF2, SDHB* and *HIF-2α* mutations.

Factors related to germline mutations such as young age (less than 40 years) with multiple tumors [11,12], positive family history [2,6,13] and presence of carotid body tumor (CBT) including bilateral presentation should be considered in HNPGLs [12,14]. Emerging evidence shows that the incidence of multiple HNPGLs and *SDHD* germline mutation positivity is relatively high in Europe. This also holds true for apparently nonfamilial cases [2,13,15,16,17,18,19,20,21,22] (Table 2). The Netherlands has shown the highest absolute prevalence of paraganglioma syndrome type 1 [20]. We discuss the phenotypic and genetic patterns in our patients with multiple HNPGLs and explore the management possibilities related to the disease.

## 2. Materials and Methods

Amongst 30 patients with HNPGLs referred to our department of otorhinolaryngology between October 2016 and March 2021 for surgical consultation, patients with multiple HNPGLs were studied. A multidisciplinary approach was adopted in all patients. After completion of clinical and radiological examination, patients were consented and referred for genetic counselling as well as analysis. Shamblin’s and Fisch’s classifications were employed to classify the extent of carotid, vagal and jugular PGLs to assess the risk of morbidity in surgical resection. Following consent for genetic testing, peripheral blood samples were taken. Genomic DNA was extracted from 10 mL of ESTA or ACD-anticoagulated blood using standardized methods. Genetic analysis was conducted using polymerase chain reaction Sanger sequencing and further next-generation sequencing (NGS) using NextSeq 500 (Illumina^®^, San Diego, CA, USA). NGS examination was used to assess 123 genes (standard panel genes for pheochromocytoma/paraganglioma). For evaluation of carrier status, index patients with positive mutation were advised to contact relatives at risk to complete genetic investigations. Finally, a treatment plan was devised in each case upon discussion with the patient. The options included surgical exploration, radiotherapy or follow-up with the intention of “wait and scan”. This depended on the age of the patient, associated comorbidities, cranial nerve status and risk of future cranial nerve palsies and genetic mutation status as well as size, localization and multiplicity of the tumor. Gender and ethnicity distributions, family history, genetic mutation status, localization and multiplicity of head and neck tumors and presence of other PGLs, as well as type of treatment, were analyzed.

## 3. Results

### 3.1. Patient Demographics and Characteristics of HNPGLs

A total of six patients (four males, two females) between the ages of 34 and 57 years were diagnosed with multiple HNPGLs. All but one were of Czech ethnicity. Patient no. 3 was of Polish origin and had a positive family history on her father’s side. Three patients had hypertension, and the youngest patient had a history of pulmonary atresia with ventricular septal defect (a rare form of tetralogy of Fallot).

Eighteen HNPGLs were found amongst six patients; three other PGLs were also diagnosed, and pheochromocytomas were absent (Table 3; Figure 1). It should be mentioned that bilateral CBTs were found in three patients (Figure 2). All tumors were of a benign character. Three patients were diagnosed incidentally with HNPGLs. Patient no. 4 was referred to us for suspicious findings on a follow-up CT scan, 5 years after extirpation of a suspected hemangioma on the left side of the neck, patient no. 5 presented to us with severe otorrhagia and the last patient in the series was diagnosed on a follow-up MRI scan for an unrelated neurological disorder. Interestingly, patient no. 2 was diagnosed with unilateral jugular tumor by other specialists on presenting with sudden right-sided facial nerve palsy in 2007. He underwent permanent embolization. He then developed a severe psychiatric illness and stopped attending his regular follow-up appointments. Finally, he returned in 2017 with severe lower cranial nerve dysfunction and very advanced disease (Figure 1A and Figure 2B). He was referred to us immediately for consultation.

### 3.2. Biochemical Test Results

All patients (including those with hypertension) had normal plasma metanephrine levels on routine reviews. Only two patients had high Chromogranin A levels, a marker for pheochromocytoma/paraganglioma (Table 4).

### 3.3. Findings of Genetic Mutation

Amongst all the patients, 50% were diagnosed with germline *SDHD* mutations and one with an *SDHB* mutation. Three out of six patients with germline mutations (67% *SDHD*-positive) had a negative family history (Table 3). All the mutations were pathogenic mutations. Patient nos. 2 and 3 with PGL1 syndrome were also diagnosed with anterior mediastinal and retroperitoneal PGLs, respectively (Figure 1A,B). A retroperitoneal PGL was also found in the *SDHB* mutation-positive patient (Figure 1C). None of the index patients’ relatives agreed to genetic counselling and testing.

### 3.4. Management Outcome

#### 3.4.1. Preoperative Embolization and Surgery for Paragangliomas

A total of three patients underwent surgery, and all of them had preoperative embolization without any complications. Patient no. 3 with PGL1 had surgical removal of the carotid body tumor and developed temporary palsy of the marginal mandibular branch of the facial nerve postoperatively, which resolved in 6 weeks. The retroperitoneal tumor was also removed successfully by other specialists. The vagal paraganglioma has remained stable. Patient no. 5 underwent two-stage surgery for the bilateral carotid (Figure 2C) and jugular tumors. He is without any complications and tympanic tumor growth has remained unchanged on regular follow-up. No new tumors were detected in these patients. The 57-year-old female patient had extirpation of her vagal tumor with no cranial nerve dysfunction, and a new tumor was diagnosed 3 years later on follow-up scans. All the tumor samples will be further examined for somatic mutation.

#### 3.4.2. Wait and Scan

The congenital heart defect in patient no. 1 was left uncorrected in childhood and led to aortic regurgitation and severe pulmonary hypertension. He was initially allocated to “wait and scan” since he was deemed medically unfit for surgery and had stable tumor growth for 3 years. However, recently, the right-sided tumor increased from Shamblin I to II (Figure 2A); therefore, he is being planned for surgery, pending medical assessment. The only patient diagnosed with PGL4 was advised surgery or radiotherapy due to the risk of malignancy. He strongly refused both, due to his previous traumatic experience during the removal of the hemangioma. Nevertheless, his disease has been stable and no signs of metastasis have been detected clinically or on radiological findings. Patient no. 2 with PGL1 and very advanced disease (Figure 2B) opted for reanimation surgery for facial nerve palsy but refused all other forms of treatment and died within 2 years from severe complications of lower cranial nerve dysfunction. All patients are clinically followed-up with a yearly MRI scan followed by a PET-CT every 3 years unless new symptoms develop earlier.

## 4. Discussion

Amongst our cohort of patients with multiple HNPGLs, 50% were diagnosed as incidentalomas, which is not an unusual phenomenon [5]. Three of our patients had *SDHD* mutations (only one had a positive family history), and one patient was diagnosed with *SDHB*. Although *SDHD* mutations can be inherited both via the maternal and paternal lines, paragangliomas almost never develop after maternal transmission of the mutation [3,23,24,25]. Maternally derived *SDHD* mutation carriers will still pass the mutation to their offspring in 50% of cases; hence, PGL1 seems to be able to skip generations. This may, in part, explain the high occurrence of *SDHD* germline mutations in apparently nonfamilial or “occult familial” cases, as seen in our patients. Paternal transmission is also associated with incomplete penetrance of 43% to 100% [26]. The *SDHD* germline mutation c.1A > G (p.Met1Val) in patient no. 2 is a missense pathogenic mutation. Interestingly, this type of mutation was reported in Germany in nonfamilial cases associated with carotid, vagal, jugular and tympanic PGLs [27] and later in a family of PGLs in China [28]. The next *SDHD* missense mutation c.112C > T(p.R38) in the Polish patient was identified amongst familial cases in the USA [29] and in an unrelated French index case with non-functional HNPGL [30]. Other large European studies reported the same mutation in HNPGLs [11,31]. The last *SDHD* splicing mutation c.53-2A > G in the eldest patient was sparingly reported by a large study in 2009 [6] and subsequently in Spanish patients [32]. Patients with an *SDHD* mutation are also commonly predisposed to CBTs (solitary or those with multiplicity) [11], thus supporting our findings of three tumors in two patients with *SDHD*. On the other hand, the *SDHB* mutation c.287G > A (p.Gly96Asp) in patient no. 4 was reported in patients with malignant catecholamine-producing paragangliomas [33] as well as in two pediatric patients with functional tumors [34]. It is difficult to precisely access malignancy in this patient, since clinical and radiographic findings did not support this. Management of multiple HNPGLs is challenging and can be categorized as (1) preventive, (2) intermediate, (3) definitive and (4) alternative.

### 4.1. Preventive

#### 4.1.1. Tumor Surveillance: Genetic Counselling, Whole Body Imaging and Biochemical Testing

Genetic counselling is the key and the first step taken towards preventative measures amongst suspected familial cases of HNPGLs. The average range of age at diagnosis of hereditary forms of *SDHD* and *SDHB* is less than 40 years [3,35]. In asymptomatic cases and incidental findings, commonly seen in HNPGLs, results may vary, as demonstrated in three of our patients. It has also become apparent that about 35% of sporadic HNPGLs are due to a germline mutation in these susceptible genes [5]. Furthermore, germline mutations in *SDHx* also occur in about 30% of HNPGLs that are regarded sporadic due to the absence of a family history [36], reiterating the theory of “occult familial” cases [37]. We reported this in our cohort of patients. Therefore, patients suspected of heritable HNPGLs should undergo genetic analysis first at the *SDHD* and *SDHB* loci, whilst if metastatic tumors or multiple abdominal paragangliomas are found without any familial presentation, the presence of *SDHB* mutations should be tested first [5]. *SDHC* is tested after exclusion of *SDHD* and *SDHB* mutations [6]. The same protocol was followed for our patients.

Genetic carrier testing should be offered to healthy first-degree relatives including second-degree relatives with *SDHD* and *SDHAF2* which are maternally imprinted. It should be started 5 years before the earliest age of onset in the family [38]. All our tumor samples will also be examined for somatic mutations since these are also detected in 30% of sporadic HNPGLs, mainly involving *VHL, NF1*, *RET* and *HIF2α* genes [5], although these are rarely associated with multiple HNPGLs.

Whole body imaging is another technique suggested for early detection; it complements genetic testing and plays a vital role in asymptomatic carriers of *SDHx* mutations [39]. Initial suggestions had been made for regular screening for development of tumors as early as 5 to 10 years of age to allow adequate treatment with minimal morbidity [21]; however, recent recommendations state testing should be started at a later age [38]. Logically, hereditary carriers of the disease should have a more detailed and frequent imaging work-up in comparison to sporadic cases [26]. In patients with multiple HNPGLs, we highly recommend annual surveillance with local anatomical imaging (MRI) and functional (PET-CT) radiological investigation every 2–3 years [21]. Identification of new tumors in multiple HNPGLs not only facilitates their detailed evaluation but also helps in changing strategies in management. Most HNPGLs are biochemically silent, as seen in our patients, and only 3–4% are catecholamine-secreting. Plasma metanephrines and methoxytyramine (a metabolite of dopamine), in particular, indicate biochemically active tumors and can be useful for monitoring such patients [40]. Chromogranin A is also released by neuroendocrine tissues along with catecholamines; nevertheless, it is a sensitive and specific diagnostic tool in detecting pheochromocytomas (familial and sporadic) rather than paragangliomas [41]. Even though two of our patients had elevated Chromogranin A levels, all tumors were otherwise biochemically silent and no pheochromocytoma was detected. Therefore, it may show a predictive risk value for future tumors.

#### 4.1.2. Malignancy in Multiple HNPGLs

Diagnosing malignancy in PGLs can be challenging and controversial, since valid histomorphologic criteria do not exist [21]. The pooled incidence for malignant paragangliomas is about 8% in *SDHD* mutation carriers, whilst for *SDHB*, it can be as high as 41% [42,43]. The highest risk of malignancy is seen in sinonasal PGLs [44]. Malignancy rates in carotid PGLs rises from 1.51% in unilateral [5] to 6–12% in bilateral cases [45]. Jugulotympanic and vagal tumors have risks of 5.1% and 6–19% [44], respectively. FDG-PET is recommended in metastatic cases with an *SDHB* mutation. Early identification of tumors is of particular importance with *SDHB* due to the risk of metastasis. PGL-related metastases occur most frequently in regional lymph nodes (nearly 70%), the bones, lungs and liver [42]. Elevated norepinephrine can be a marker for metastatic PGLs [42]. Furthermore, Chromogranin A does not provide an additive benefit to standard surveillance for predicting the presence of SDHB- or SDHD-related paraganglioma but has a useful negative predictive value when normal in patients with an SDHB mutation [46]. Suggestions such as metastasis to non-neuroendocrine tissues [47], *SDHB* mutation carriers and tumors greater than 5 cm in diameter [41] can be significant predictive factors for malignancy.

These techniques have certain disadvantages as well. Noncompliance with genetic testing increases the possibilities of underestimating the risk of tumor development in asymptomatic carriers. It also does not offer antenatal or in vitro mutation analysis for panel genes of pheochromocytoma/paraganglioma, hence creating uncertainty amongst young syndromic patients who decide on parenthood. This would be beneficial to at least 50% of our patients.

### 4.2. Intermediate

#### Wait and Scan Approach

HNPGLs are mostly benign and very slow-growing in nature (1–2 mm/year) [48], and thus expected growth would be from 1 to 2 cm in 10 years’ time. Therefore, it follows that the character of these tumors can be studied over time and the “wait and scan” policy can be used for asymptomatic cases with a low risk of malignancy. Nonetheless, this approach should be used with caution in multiple HNPGLs, especially with a germline mutation; this can lead to destruction of adjacent structures and irreversible complications. It has been demonstrated that this approach could not prevent tumor-induced complications in 16% of nongrowing tumors [49].

### 4.3. Definitive

#### Surgical Therapy and Radiotherapy

The main aim is achieving long-term tumor control with minimal cranial nerve morbidity. The risk of major vascular injury is high for CBTs, especially for Shamblin III; however, for vagal tumors, it could be as high as 100% [26]. It is recommended that patients should undergo preoperative embolization to reduce perioperative morbidity [45]. Long-term cure rates after complete surgical resection of HNPGLs have been reported between 90% and 100% in carotid, vagal and tympanic tumors [50,51,52]. In jugular tumors, cure rates have been achieved in up to 72.8% of cases but with high cranial nerve palsies [50]. This would be even more difficult to achieve in multiple HNPGLs; therefore, surgery should never be performed in a single stage, thus avoiding bilateral cranial nerve deficits and irreversible disabilities [26]. It is recommended to remove the larger tumor first, and then the contralateral side can be managed in combination with either other approaches or surgery. We used the same policy in our patients. In patients with jugular tumors, a surgical interval of 9 to 12 months should be maintained, since the jugular bulb is usually resected, and venous collaterals need to be developed. Alternatively, preoperative radiotherapy may be used in jugular PGLs, where a high risk of cranial nerve injuries is expected, and there is a possibility of incomplete resection and/or an aggressive behavior of these tumors [26]. Since paragangliomas are radiosensitive, radiotherapy is the second most commonly used technique. Radiosurgery (image-guided radiosurgery or stereotactic surgery) uses a more precise form of therapeutic radiation and almost eliminates side effects seen with conventional radiotherapy. Furthermore, actuarial 10-year progression-free survival can be above 90% [53,54]. This approach could be beneficial with large jugular tumors, where some degree of vagus nerve dysfunction exists preoperatively [55] or has a relatively high risk of lower cranial nerve deficit postoperatively [56]. Although comparative analysis showed clinical improvement in patients who underwent Gamma Knife radiosurgery versus microsurgical resection in jugular tumors [57], cranial dysfunction, pre-existing with large tumors (of more than 7 cm) or postoperatively, still has a risk of worsening the symptoms [54]. Nonetheless, the higher risk of cranial nerve surgery with open surgery [58] and an operative mortality of 1 in 100 supports the use of this technique even as a frontline modality [50]. In multiple tumors, this might be of significant value since it reduces the risk of severe debilitation. Radiosurgery would be considered in at least two of our postoperative patients for the other paragangliomas if tumor growth was noted on follow-up scans. The long-term risk of developing delayed radiation-induced malignancies when treating benign paragangliomas, which have been reported up to 15 years after radiation [53], should be considered in young patients with multiple tumors.

### 4.4. Alternative

#### Targeted Therapy, Radionuclide Therapy and Therapeutic Radiation

In order to understand the concept of targeted therapy, it is important to understand the mechanism of tumorigenesis in HNPGLs. Based on transcriptomes, tumorigenic pathways involved in the development of HNPGLs can be divided into two main clusters (Table 1). In cluster 1 tumors, glycolysis is activated [59], and angiogenesis as well as hypoxia is increased [4]. Meanwhile, cluster 2-related tumors include genes that mediate translation initiation, protein synthesis, adrenergic metabolism, neural/neuroendocrine differentiation and abnormal activation of kinase signaling pathways such as RAS/RAF/MAPK and PI3K/AKT/mTOR [5,59]. Therefore, pathogenic factors associated with these tumors can be targeted accordingly. Sunitinib, a tyrosine kinase inhibitor that targets vascular endothelial growth factors which inhibit angiogenesis, has shown varying but not very encouraging results even amongst patients with metastasis. Many other possibilities have also been discussed [4].

Radionuclide therapy such as somatostatin analogues (^177^Lu-DOTATATE) has been demonstrated in four patients with non-metastatic non-resectable progressive PGL1 syndrome, where a partial response and disease stability were achieved [60]. The use of therapeutic radiation to treat multifocal tumors also brings a promising future, and the possibility of treating multiple sites in one to three outpatient sessions by the use of a combination of antiangiogenics and radiosurgery. This would ideally lead to the disruption of the neovasculature and the tumor’s supply of oxygen and nutrients. Antiangiogenic therapy has to be administered prior to radiosurgery, but the optimal time to initiate this antiangiogenic therapy for radiosensitization in a patient is yet to be determined [26]. However, the effectiveness of using such therapeutic options is still debatable.

## 5. Conclusions

We found four pathogenic mutations amongst six patients with multiple HNPGLs that are not so commonly seen. These tumors were of a benign character, even in the patient with an *SDHB* mutation. Management of patients with multiple HNPGLs should be geared towards cure or tumor growth control and always with a multidisciplinary approach. Preventative measures represent the gold standard in effectively controlling the disease in index patients and their relatives but require patient compliance. In vitro and prenatal testing for panel genes of pheochromocytoma/paragangliomas may bring new insights to the disease and help reduce the risk of development of the disease. However, since the disease shows incomplete penetrance and the risk of developing tumors is less than 50%, this may raise ethical dilemmas. Presently, the mainstay of treatment includes surgery and/or radiotherapy. Combination therapy should be used in multiple HNPGLs where indicated, in order to reduce the risk of morbidity. Whilst molecular examination predicts the phenotype, inheritance and the risk of development of a tumor, it cannot be used to delineate tailored therapy according to mutation type. Therefore, other modalities such as gene targeted therapy although show a massive potential due to the versatile tumorigenic pathways of the disease, it is of little practical use in current times. Patients with germline mutations and multiple tumors should be followed up very closely, more so in *SDHB* mutation tumors due to the high risk of malignancy. Definitive algorithms for clinicians should be adopted for centers where HNPGL patients are managed.

## Figures and Tables

**Figure 1 biomedicines-09-00626-f001:**
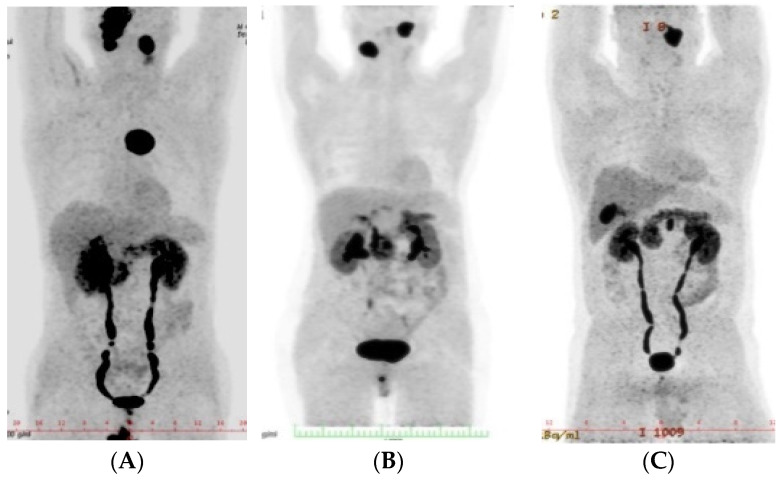
Whole body imaging using 18F-FDOPA PET-CT demonstrating multifocal paragangliomas in (**A**) patient no. 2, (**B**) patient no. 3 and (**C**) patient no. 4.

**Figure 2 biomedicines-09-00626-f002:**
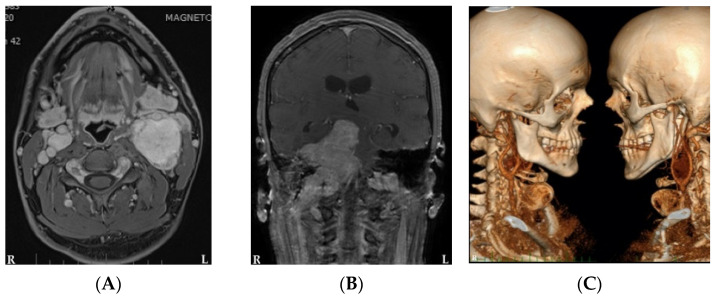
Bilateral CBTs in patients with multiple HNPGLs: (**A**) MRI of the neck (axial view) in patient no. 1 with congenital heart disease and absence of germline mutation; (**B**) MRI of the neck (coronal view) in patient no. 2 with advanced disease and *SDHD* mutation; (**C**) 3D reconstruction of CT angiography neck (lateral views) in patient no. 5 with absence of germline mutation.

**Table 1 biomedicines-09-00626-t001:** Summary of genes with mutations related to head and neck paragangliomas (reproduced with permission from Guha, A.; Musil, Z.; Vicha, A.; Zelinka, T.; Pacak, K.; Astl, J.; Chovanec, M. A Systematic Review on the Genetic Analysis of Paragangliomas: Primarily Focused on Head and Neck Paragangliomas. *Neoplasma* 2019, *66* (5), 671–680) [5].

		Cluster 1							Cluster 2		
Gene		SDHD	SDHAF2	SDHC	SDHB	SDHA	VHL	HIF-2α	RET	NF1	TMEM127
Locus		11q.23	11q13.1	1q21	1p36.13	5p15.33	3p25.3	2p21-p16	10q11.2	17q11.2	2q11.2
Protein function		Structural subunit of the mitochondrial protein complex II (SDH)	Mitochondrial assembly factor for complex II	Structural subunit of the mitochondrial protein complex II (SDH)	Core subunit of the mitochondrial protein complex II (SDH)	Core subunit of the mitochondrial protein complex II (SDH)	Regulates HIF1a and HIF2a proteasomal degradation	Transcription factor of the bHLH-PAS protein family	Transmembrane tyrosine kinase receptor for extracellular signal molecules of the GDNF family	Inhibits the GTPase HRAS and disrupts the RAS signaling pathway	Probable role in endosomal trafficking and mTOR regulation
Syndrome		PGL1	PGL2	PGL3	PGL4	PGL5	VHL	Paraganglioma- somatostatinoma-polycythemia	Sipple	NF1	NA
OMIM ID		168000	60650	605373	115310	614165	193300	611783	171400	162200	613903
Inheritance		AD PI	AD PI	AD	AD	AD	AD/Somatic	Somatic	AD PI	AD	AD
HNPGL		High	High	Medium	Medium	Low	Very low	Very low	Very low	Very low	Very low
Other PGLs		Medium	NA	Low	High	Low	Low	Medium	NA	NA	Variable
Multiple PGLs		High	Medium	Low	Medium	NA	Variable	Medium	None	None	None
Associated PHEOs		Low	None	Variable	Medium	None	High	Low	Medium	Low	High
Malignancy risk		Low	NA	Low	High	NA	Low	NA	Low	High	Low
Relative mutation frequency	Germline	High	Low	Medium	High	Medium	High	Low	High	Medium	Low
	Somatic	NA	NA	Low	High	NA	High	High	High	Very high	NA
Other features		GIST, rarely papillary thyroid cancer	GIST	GIST	GIST rarely renal cell cancer	NA	CNS and eye hemangioblastomas, clear cell renal cancer, islet cell tumor	Somatostatinoma, polycythemia	Medullary thyroid cancer, pituitary adenoma	Café-au-lait spots, Lisch nodules, fibromatous skin tumors	NA

AD, autosomal dominant; PI, paternal inheritance; HNPGL, head and neck paraganglioma; PGL, paraganglioma; PHEO, pheochromocytoma; NA, not available; CNS, central nervous system; GIST, gastrointestinal stromal tumor; bHLH-PAS, basic helix-loop-helix-PER-ARNT-SIM; GDNF, glial cell line-derived neurotrophic factor; mTOR, mammalian target of rapamycin.

**Table 2 biomedicines-09-00626-t002:** *SDHD* mutation in multiple head and neck paragangliomas.

References/Authors	Country of Study	Duration of Study (Years)	No. of Cases with Multiple HNPGLs	No. with Germline *SDHD* Mutations (%)
Dannenberg et al. 2002 [15]	Netherlands	12	17 (familial)	17 (100%)
			10 (isolated)	7 (70%)
Astuti et al. 2003 [16]	United Kingdom	9	1 (familial)	1 (100%)
			3 (isolated)	3 (100%)
Badenhop et al. 2004 [17]	Australia	10	11 (familial)	9 (82%)
Lima et al. 2007 [18]	Spain	24	4 (familial)	3 (75%)
Fakhry at al. 2008 [13]	France	13	4 (familial)	4 (100%)
			3 (isolated)	2 (67%)
Persu at al. 2008 [19]	Belgium	3	12 (familial)	10 (83%)
Hensen et al. 2011 [20]	Netherlands	59	173 (familial)	130 (75%)
			22 (isolated)	14 (64%)
Papaspyrou et al. 2011 [21]	Germany	21	22 (unspecified)	19 (86%)
Piccini et al. 2012 [2]	Italy	8	4 (familial)	4 (100%)
			10 (isolated)	10 (100%)
Shulskaya et al. 2018 [22]	Russia	Unspecified	8 (isolated)	5 (63%)
Present study 2021	Czech Republic	4.5	1 (familial)	1 (100%)
			5 (isolated)	2 (40%)

**Table 3 biomedicines-09-00626-t003:** Characteristics of patients with multiple HNPGLs.

Patient No.	Age at Diagnosis (Years)	Gender	Genetic Mutation	Syndrome	Type and Localization of HNPGLs	Classification of Tumors	Other PGLs
1	34	M	-	-	Carotid (B)	L: Shamblin III, R: Shamblin II	-
2	36	M	*SDHD:* c.1A > G (p.Met1Val)	PGL1	Carotid (B) Vagal (B) Jugular (R)	Shamblin III	Anterior
						Fisch C	Mediastinum
						Fisch C4 Di2	
3	43	F	*SDHD:* c.112C > T,p.R38	PGL1	Carotid (R) Vagal (L)	Shamblin II	Retroperiotoneal
						Fisch A	
4	47	M	*SDHB:* c.287G > A (p.Gly96Asp)	PGL4	Vagal (L) Jugular (L)	Fisch A	Retroperiotoneal
						Fisch C1	
5	51	M	-	-	Carotid (B) Jugular (B) Tympanic (L)	Shamblin II	-
						Fisch C1	
						Fisch A1	
6	57	F	*SDHD:* c.53–2A > G	PGL1	Vagal (R) Jugular (R)	Fisch A	-
						Fisch C1	

*SDHD:* succinate dehydrogenase complex subunit D; *SDHB:* succinate dehydrogenase complex subunit B; L: left; R: right; B: bilateral.

**Table 4 biomedicines-09-00626-t004:** Biochemical results of patients with multiple HNPGLs.

Patient No.	Plasma Metanephrine (0.140–0.540) nmol/L	Plasma Normetanephrine (0.130–0.790) nmol/L	ChromograninA (0–85) ng/mL
1	0.063	0.308	32.7
2	0.012	0.186	231.4
3	0.171	0.342	224.7
4	0.102	1.306	72.6
5	0.186	0.231	22.5
6	0.031	0.285	78.2

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
