# Peer review of "Genetic Variants in Patients with Multiple Head and Neck Paragangliomas: Dilemma in Management"

_biomedicines, 2021, doi:10.3390/biomedicines9060626_

Round 1

Reviewer 1 Report

It is a well conducted work, however, there is somewhat lack of relevance in the "conclusions".

There is a low numbers on these tumors to stablish any common clinical management; specially if we split in many molecular subgroups with even lower number of patients.

Molecular determinants does no seem to have a critical influence in the managements. The discusion around what may supose biologically these mutations is good though.

There are specific therapies applied according to these mutations? How these tumors respond to these therapies. In most if not all mutation there are not therapies against these mutations, The will change critically the management.

It is a good work, but at the end it is a good review of  paraganglioma.

Author Response

Thank you very much for reviewing and supporting the proposed paper. We hereby clarify some of the issues raised by you.

We demonstrated 18 HNPGLs and 3 other paragangliomas amongst 6 patients within 4.5 years, which is comparatively higher to some other countries (please kindly refer to Table 2 in the manuscript). In current times, molecular determinants do not have a great influence on management, that is, there is a lack of precision therapy in terms of type of mutation. Therefore, although cluster differentiation is seen amongst the different types of mutation and modalities such as targeted therapy are based on these pathways, no evidence has surfaced on the efficacy of such therapy based on specific mutation type. A general overview of the management would be appropriate here since in our cohort, we also have patients with progressive disease who are unable to undergo or refused surgery. Case-based or tailor-made therapy would be ideal in patients with germline mutations, but unfortunately this will require further research.

Minor changes have been made to the conclusion of the manuscript as recommended by you (please see section ‘5. conclusions’).

Furthermore, English language editions have been done accordingly and reflected throughout the manuscript (relevant sections are marked in red color).

Reviewer 2 Report

Nice work

Author Response

Thank you so very much for reviewing and supporting the proposed paper. English language editions have been done accordingly and reflected throughout the manuscript (relevant sections are marked in red color).

We would also like to kindly bring to your attention, that minor changes have been made to the conclusion of the manuscript as recommended by another reviewer (please see section ‘5. conclusions’).

Reviewer 3 Report

The manuscript “Genetic Variants in Patients with Multiple Head and Neck Paragangliomas: Dilemma in Management”

By Anasuya Guha, Ales Vicha, Tomas Zelinka, Zdenek Musil and Martin Chovanec

evaluated the genetic mutations in the standard susceptibility genes for pheocromocytoma/paraganglioma of 6 patients affected by multiple primary head and neck paragangliomas. The analyses of clinical, biochemical and genetics features of patients with HNPGLs are well carried out. The patients follow up was 4.5 years. The management outcomes and appropriate care and needs of the patients affected by multiple primary HNPGLs are suggested.The manuscript is well written, questions are well posed and the methods appropriate.

A typo to correct is the following:

Lane 186: “this.nManagement” correct to “this management”

Author Response

Thank you so very much for reviewing and supporting the proposed paper.

The typo has been corrected as suggested in your comments. Furthermore, English language editions have been done accordingly and reflected throughout the manuscript (relevant sections are marked in red color).

We would also like to kindly bring to your attention, that minor changes have been made to the conclusion of the manuscript as recommended by another reviewer (please see section ‘5. conclusions’).